# Umami in Wine: Impact of Glutamate Concentration and Contact with Lees on the Sensory Profile of Italian White Wines

Deborah Franceschi [ID], Giovanna Lomolino *, Ryo Sato, Simone Vincenzi and Alberto De Iseppi [ID]

Department of Agronomy, Food, Natural Resources, Animal and Environment DAFNAE, University of Padua, Viale dell'Università 16, 35020 Legnaro, Padova, Italy
* Correspondence: giovanna.lomolino@unipd.it; Tel.: +39-(0)498272917; Fax: +39-(0)498272919

**Abstract:** Umami is a fundamental taste, associated with the molecules of monosodium glutamate and other amino acids and nucleotides present in many fermented foods and beverages, including wine. Umami also plays the role of flavor enhancer and prolongs the aftertaste. In this research, monosodium glutamate and aspartate, responsible for the umami taste, were quantified in Italian still and sparkling white wines aged through contact with yeasts. The wines were studied from a sensory point of view to quantify the perception of umami and relate it to other sensory parameters. The results show that monosodium glutamate and aspartate are present in the wines studied. However, sensory analysis shows that there is no clear relationship between the umami taste and the concentration of the two amino acids, but their presence plays a fundamental role in enhancing other gustatory and olfactory perceptions, making them even more persistent.

**Keywords:** umami perception; white wines; aging on lees; yeast peptides; sensory profile; taste perception

## 1. Introduction

The Japanese term "Umami" is used all over the world to indicate a pleasant taste sensation that is qualitatively different from the other primary tastes: sweet, salty, bitter, and sour [1]. Even if there are no equivalents in other languages, this fifth taste is generally translated as "meaty", "savory", and "tasty".

Foods rich in compounds attributable to the umami taste are present in different food cultures (chicken broth, meat extract, fermented food and sausages, and seasoned cheeses) [2]. However, consumers find it difficult to identify this taste in food [3,4]. Indeed, umami is not a prominent taste, but it is essential to obtain harmony among other tastes as it gives the sensation of fullness, robustness and, in general, satisfaction that makes it one of the critical factors determining the palatability of foods [5]. As the other primary tastes, umami sensation is induced by the interaction between well-defined compounds and specific taste receptors in the mouth [6]. The main molecule responsible for the umami taste is monosodium glutamate, the sodium salt of glutamic acid, one of the most abundant amino acids in foods [2]. Even if monosodium glutamate is naturally present in many food preparations, it can be also added as a taste enhancer in place of table salt [3]. Other molecules associated with the umami taste are aspartic acid, another amino acid with a taste intensity that is about 7% of glutamate [2]; inosinate (IMP) and guanylate (GMP), which derive from the degradation of nucleic acids [7].

Even if the umami taste is commonly associated with foods rich in glutamate, amino acids and nucleotides, such as fermented animal products, this sensation can play a significant role in defining the sensory profile of multiple fermented products including wine [3,5]. In particular, various experiments [5] have demonstrated the presence of glutamic acid and glutamate in wines, even if their concentrations were below the perception threshold [8].

The profile of amino acids and peptides in wine depends on the grape variety, vintage, degree of ripeness, geographical origin, fermentation, and aging conditions [9]. However,

glutamic acid is one of the main free amino acids in *Vitis vinifera*, mainly located in the skins of grape berries [10]. Even if its concentration decreases during alcoholic fermentation, when all amino acids are used by yeasts as nitrogen sources, glutamic acid remains one of the most abundant free amino acids in wine [9]. The wine's amino acid profile is also affected by the different oenological practices.

For example, sweet wines, produced by stopping fermentation or adding alcohol, are characterized by a higher content of amino acids compared to dry wines. For this reason, high glutamate content is found in fortified wines such as Madeira, Sherry, and Port [5].

Additionally, the maceration time can affect the presence of glutamic and aspartic acid in wine which is shown to decrease, at least in the first 12 h [11]. Interestingly, in another research, the application of thermal or PEF treatments during the maceration on Parellada grapes showed no significant effect on the content of both amino acids [12]. Conversely, the application of a carbonic maceration on Agiorgitiko red grapes showed an increase (+48%) of aspartic acid and a parallel decrease (−38%) of glutamic acid [13].

Additionally, in Italian *passito* ("withered") wines, a moderate gradual increase in glutamic acid and other amino acids during the drying process has been observed in Corvina grapes and attributed to a possible degradation of existing proteins by proteases and peptidases released from damaged cells [14].

However, the main contribution on the accumulation of umami-related amino acids is commonly attributed to the time of contact with lees [5,15]. Wine lees consists of dead yeast cells which, after several weeks of autolysis (enzymatic degradation), release nitrogenous compounds, polysaccharides, and glycoproteins into the wines. However, this aspect has been poorly studied and the results are difficult to interpret due to multiple factors including the release of yeast peptides [16]. Some wines are left "*sur lies*" for their ageing. An example is the production of sparkling wines with the classic method, where for the second fermentation the yeast lees remain in the wine bottle for a minimum of 15 months. This prolonged contact of wine with autolyzed yeasts leads to an overall increase in amino acids. Champagnes, which are aged for 5 years or more with their lees, report a higher glutamate content compared to sparkling wines produced in an autoclave with the Charmat/Martinotti method [5,15].

In other wines aged on lees, the autolysis can be accelerated by the resuspension of the lees in the wine tank (*batonnage*) through which the amount of amino acids increases by more than 60% [5]. Nevertheless, even in these wines, the umami taste is not included among the sensory descriptors as the concentrations of glutamate are known to be below the human detection threshold (0.3 g/L) [6,17]. However, as umami-related molecules also known to elicit other flavors in foods, it is possible to hypothesize that, even when they are present in low concentration, these molecules can affect the intensity of other taste attributes thus changing the overall sensory perception of wines. A similar behavior has already been noted in other foods [18].

Since the umami taste is an object of attention, as it affects the sensory perception of fermented beverages, there is the need to better study its perception and interaction with other sensory parameters. Furthermore, it is also important for wine producers to study the effects of the compounds responsible for this taste in order to implement viticulture and winemaking processes that will allow for a certain concentration of umami precursors.

Therefore, the purpose of this work was, initially, to train the panel to recognize the umami taste and verify its perception in a neutral wine with the addition of monosodium glutamate and verify the possible variation of the other tastes. Some wines, still and sparkling, obtained in the production area of wine DOCG di Conegliano and Valdobbiadene will also be analyzed, to verify how the content of the molecules responsible for the umami taste varies, based on the contact times with the yeasts in their production process and relate them to the results of the sensory profile.

## 2. Material and Methods

### 2.1. Experimental Design

The sensory analysis was carried out over 3 parts, each divided into several sessions. In the first part, the panel was trained on umami taste recognition. At first, they were asked to recognize different gustatory perceptions and the mouthfeel (tactile perception), intended as the synergy among acidity, tannins, alcohol, glycerol, $CO_2$ and other constituents also perceived at a tactile level. Then, in the same session, a test to recognize the threshold of umami and salty tastes was also submitted, using different concentrations of sodium glutamate and sodium chloride in water (Sections 2.3 and 2.4).

The second part (carried out in three sessions) involved a recognition test of the umami perception threshold, like the previous part, but on a commercial neutral white wine (Tavernello, Caviro, Italy) tasted alone and after the addition of pure monosodium glutamate (MSG) (Section 2.5).

In the third part, a quantitative descriptive analysis (QDA) including different umami taste indicators was performed on the 3 wine samples of the second part (Tavernello wine added with 0, 0.05, 0.15 g/L of MSG) along with 6 commercial white wines produced with a different period of contact with yeast lees. For all these 9 wines the concentration of monosodium glutamate and aspartate (ASP) was analytically determined by HPLC (Section 2.6).

In order to find the correlation between the different variables (time of contact with lees, vintages, MSG concentration, sensory indicators), data were analyzed with multivariate statistical analysis (Section 2.7).

### 2.2. Sensory Panel

All sensory analyses were conducted by the Panel of the Interdepartmental Center for Research in Viticulture and Oenology (CIRVE) of the University of Padua in Conegliano (Treviso, Italy), which was composed of 9 judges, 5 male and 4 female, aged between 30–50 years old. The analytical sensory panel, made up of 9 judges, is represented by people highly trained for years on the evaluation of wine and the study of its parameters. It is regularly trained according to ISO standards and is the result of a selection that took place years ago.

The training was repeated seven times in order to familiarize the judges with the descriptors and with the umami taste in particular. Panelists were trained for general (aromatic intensity, taste intensity, sour, sweet, mouthfeel, taste persistence, salty) and umami-related (yeast smell, umami) sensory descriptors. For training on yeast smell, bakery yeast was used. Other standards were chosen among the ones indicated in the ISO 3972:2011 [17]. The panel training was carried out in two phases: gustatory-tactile recognition test, and perception and recognition threshold test, according to ISO 8586:2012 [19]. The training made it possible to obtain a panel of 9 judges prepared for the recognition of the descriptors without further selections. For tests involving wine samples, the OIV's review on the sensory analysis of wine [20] was used.

All the panel sessions were conducted in a professional sensory analysis room, at room temperature (21 ± 0.5 °C) and with artificial light.

### 2.3. Gustatory-Tactile Recognition Test

During the first session, the judges had to distinguish six perceptions (sweet, salty, sour, bitter, and umami and mouthfeel) in eight standards (eight different concentrations for each standard parameter associated with each perception) according to ISO 8586:2012 [19]. In brief, six pure compounds related to different gustatory or tactile sensation [19] have been dissolved into six glasses of water coded with three random numbers (three-digit code) to anonymize them while the remaining two glasses contained only water (blanks).

For this test, Lauretana (Graglia, Italy) natural mineral water was chosen for its low concentration of sodium cation (0.88 mg/L). To decide the concentration of the solutions for training the panel, the value of the identification test indicated by the ISO 8586:2012 [19]

standard was considered. The judges were experts; therefore, for the four primary tastes, sweet, salty, sour, and bitter, slightly lower concentrations than those required by ISO 8586:2012 [19] were used for training the judges in taste recognition (Table 1). On the contrary, since the umami taste was the core of this work, to make it well recognized by the judges, a double concentration of monosodium glutamate (MSG) was set compared to that suggested by ISO 8586:2012 [19].

**Table 1.** Concentrations of the solutions used for the recognition threshold test.

| Taste | Chemical Compound | Concentrazion (g/L) | | | | |
|---|---|---|---|---|---|---|
| Salty | Sodium cloride | 0.25 | 0.5 | 1.0 | 1.5 | 2 |
| Umami | Monosodium glutamate | 0.15 | 0.3 | 0.6 | 0.9 | 1.2 |

The presence of two samples, containing only water, prevented the judges from adopting biased behavior, as suggested by ISO 8586:2012 [19].

After compilation, the judges were provided with the correct answers, then re-tasted the samples to fix the information.

*2.4. Umami Recognition Threshold in Water*

The objective of the test was to verify whether the judges recognized the umami taste and, subsequently, the concentration of their perception threshold. To carry out this test, two series of solutions were prepared: the first with sodium chloride and the second with monosodium glutamate at different concentrations. The solutions were prepared with five dilutions, starting from the concentration for the identification of tastes of ISO 3972:2011 [17] (Table 1). The judges had to taste the five solutions of each series, presented randomly.

The judges were asked to indicate the solution in which they perceived the tastes and recognized them. The minimum value at which 50% of the judges identified a taste is the perception threshold [17].

*2.5. Umami Recognition Threshold in Wine*

As the umami perception threshold can be influenced by various components in wine [13], the second sensory analysis session was performed, switching the matrix from water to a neutral table white wine (Tavernello Bianco; Caviro S.p.A., Faenza, Italy).

Therefore, the solutions were prepared by dissolving monosodium glutamate in the neutral white wine at the concentrations reported in Table 2. Considering that glutamate is a molecule already present in wine, lower quantities (0.05, 0.15 g/L) of MSG have been added compared to the previous recognition threshold (in water).

**Table 2.** Monosodium glutamate (MSG) concentrations used for the threshold test of umami perceptions in Tavernello wine (Tav).

| Taste | Chemical Compound | Concentration (g/L) | | |
|---|---|---|---|---|
| Umami | Monosodium glutamate | 0 | 0.05 | 0.15 |
| | Sample name | Tav | Tav0.05MSG | Tav0.15MSG |

Each judge had to evaluate three samples, coded with three-digit numbers and randomized. All parameters were assessed using the 100-point scale according to the International Union of Oenologists IUOE [21] The judges had to assess whether they perceived the umami taste in the samples and had to indicate its intensity on the structured scale (from 0 to 100).

*2.6. Quantitative Descriptive Analysis of the Different White Wines*

In this third sensory session, three types of wine were examined (Table 3): (i) the neutral table wine Tavernello alone and after the addition of MSG as described in Section 2.5; (ii) a sparkling white wine: Bianco Colli Trevigiani IGT "Vin Col Fondo" (Lucchetta winery, Conegliano, Italy), which was naturally refermented in the bottle and stored on the lees in three consecutive vintages (2019, 2020, and 2021); (iii) a still white wine "Bianco Colli di Conegliano DOCG" (Lucchetta winery, Conegliano, Italy), produced with the *batonnage* technique for different times. Additionally, these samples came from three consecutive vintages (2019, 2020, and 2021).

**Table 3.** Description of the white wines.

| Wine and Denomination | Year | Grape Variety | Resting Period on the Lees | Abbreviation |
|---|---|---|---|---|
| Bianco Colli di Conegliano DOCG (not bottled) | 2021 | 50% Manzoni bianco 40% Chardonnay 5% Riesling 5% Sauvignon | Batonnage for 2 months | WB21 |
| Bianco Colli di Conegliano DOCG (not bottled) | 2020 | | Batonnage for 16 months | WB20 |
| Bianco Colli di Conegliano DOCG (bottled) | 2019 | | Batonnage for 12 months | WB19 |
| Vino Bianco Colli Trevigiani IGT "Vin Col Fondo" (still wine) | 2021 | Glera 100% | No | Pros still21 |
| Vino bianco Colli Trevigiani IGT "Vin Col Fondo" (sparkling) | 2020 | | 1 year | Pros spark20 |
| Vino bianco Colli Trevigiani IGT "Vin Col Fondo" (sparkling) | 2019 | | 2 year | Pros spark19 |

In another distinct session, panelists assessed the wines with the only parameter of "overall liking" in a points test with a structured scale (0–100).

In this session, using points test with a structured scale (0–100), the panel described the wine according to the following indicators: umami, sweet, yeast smell (the smell of commercial bakery yeast), salty, sour, taste intensity, aromatic intensity, mouthfeel, taste persistence (defined as the permanence of olfactory-gustatory perceptions over time and is measured in seconds).

*2.7. HPLC Analysis*

The analysis of glutamate content in wines was determined on a HPLC (Nexera XR, Shimadzu, Milan, Italy) equipped with a fluorescent detector (RF-20A XS, Shimadzu). The samples were derivatized using the AccQtag kit (Waters, Milford, MA, USA) for amino acid determination. Briefly, 10 µL of wine sample were mixed with 70 µL of AccQTag Borate buffer and 20 µL of AccQTag reagent (6-aminoquinolyl-N-hydroxysuccinimidyl carbamate) and incubated at 55 °C for 10 min. Five µL of the solution were injected in HPLC and separated on the Nova-Pak C18 column included in the kit. The separation was performed at 37 °C, using a gradient from Waters AccQTag Eluent A to acetonitrile, as reported in the Waters AccQTag manual. The flow rate was set at 1 mL/min. The peaks were detected with 250 nm of excitation wavelength and 395 nm of emission wavelength. The identification and quantification of different amino acids was obtained running in the same condition the calibration standard provided with the kit.

*2.8. Statistical Analysis*

Data were statistically processed by Excel (Microsoft Corporation, Redmond, WA, USA), Statgraphics Centurion XVI (StatPoint Technologies Inc., Warrenton, VA, USA), and OriginPro (OriginLab Corporation, Northampton, MA, USA). A descriptive statistical

study, spider plots (sensory parameters) was conducted. Analysis of variance (ANOVA) followed by Tukey's test, and HSD were applied for inferential study ($p < 0.05$). Principal component analysis (PCA) and multiple linear regression were applied to sensory and analytical parameters.

## 3. Results and Discussion

Glutamic acid, the amino acid responsible for the umami taste, accumulates in fermented food and beverages due to yeasts' autolysis and proteolytic activity. In the case of wine, the presence of glutamic acid is dependent on the time and methods of contact with lees, and by the amount of proteins initially present in the wine which, in turn, depends on the grape variety [22,23]. In addition to having its own taste, umami influences the perceptions of other tastes, in some cases enhancing them, as in the case of salty and sweet, in others suppressing them, as in the case of acidity and bitterness [24].

### 3.1. Umami Recognition Thresholds

In this research, 60% of panelists recognized umami taste at the concentration of 0.6 g/L, which is higher than what was reported by Schmidt et al. (2021) who identified the threshold on the value 0.3 g/L in aqueous solution.

Subsequently, the umami taste was identified in wine, a more complex matrix than water, which could interfere with the perception of taste. Interestingly, 50% of panelists recognized this taste in neutral wine (Tav), where the detected MSG concentration was 0.048 g/L. The recognition of the umami taste in wine, despite the low concentration, could be due to the presence of other umami related molecules responsible for gustatory perception, or they could act in synergy, promoting its recognition [6].

In the other Tav wine samples, added with MSG, 100% of the panelists recognized the umami taste when present in concentrations of 0.097 g/L in Tav0.05MSG and 0.181 g/L in Tav0.15MSG, respectively.

### 3.2. QDA (Qualitative and Descriptive Analysis) of Wine

As shown in Figure 1A, the perception of umami is present and identified in all 3 wine samples. As expected, the umami taste was perceived with higher scores in the Tav0.15MSG sample compared to Tav and Tav0.05MSG samples which, interestingly, reported a comparable intensity score. Furthermore, the typical umami descriptors: salty, yeast smell, along with, taste persistence and intensity, were better perceived in Tav0.05MSG and Tav0.15MSG than Tav, confirming how MSG acts as an enhancer of sensory perceptions. These data confirmed those of [25], that reported that umami taste acts as a flavor enhancer for other gustatory perceptions, such as sweet and salty, even when the umami itself is below its threshold of perception [5,23].

Even if it is odorless, MSG seems also to improve the aromatic perceptions of wine. A similar effect was noted in a previous study were the increase in odor intensity in cheese was promoted by the presence of MSG, even if the umami taste directly associated with the molecule was not perceived [26].

It is known that oenological practices are responsible for the sensory characteristics of wines, such as for example the *batonnage* practice, which consist in keeping yeast lees in suspension by stirring the wine [27], favoring the release of nitrogenous compounds, polysaccharides and glycoproteins from yeast cells. With this winemaking technique, the release of amino acids, including glutamic acid, can increase by 60% [5]. From a sensory point of view, the wine aged with this technique is well-balanced and smooth [10]. For this reason, 3 wines "Bianco Colli di Conegliano DOCG", obtained using the *batonnage* technique, were analyzed by the sensory profile to compare the perception of umami taste and to study the other sensory parameters associated with this taste. In addition, the effect of three vintages was evaluated: 2019, 2020, and 2021 (WB19, WB20, and WB21 samples).

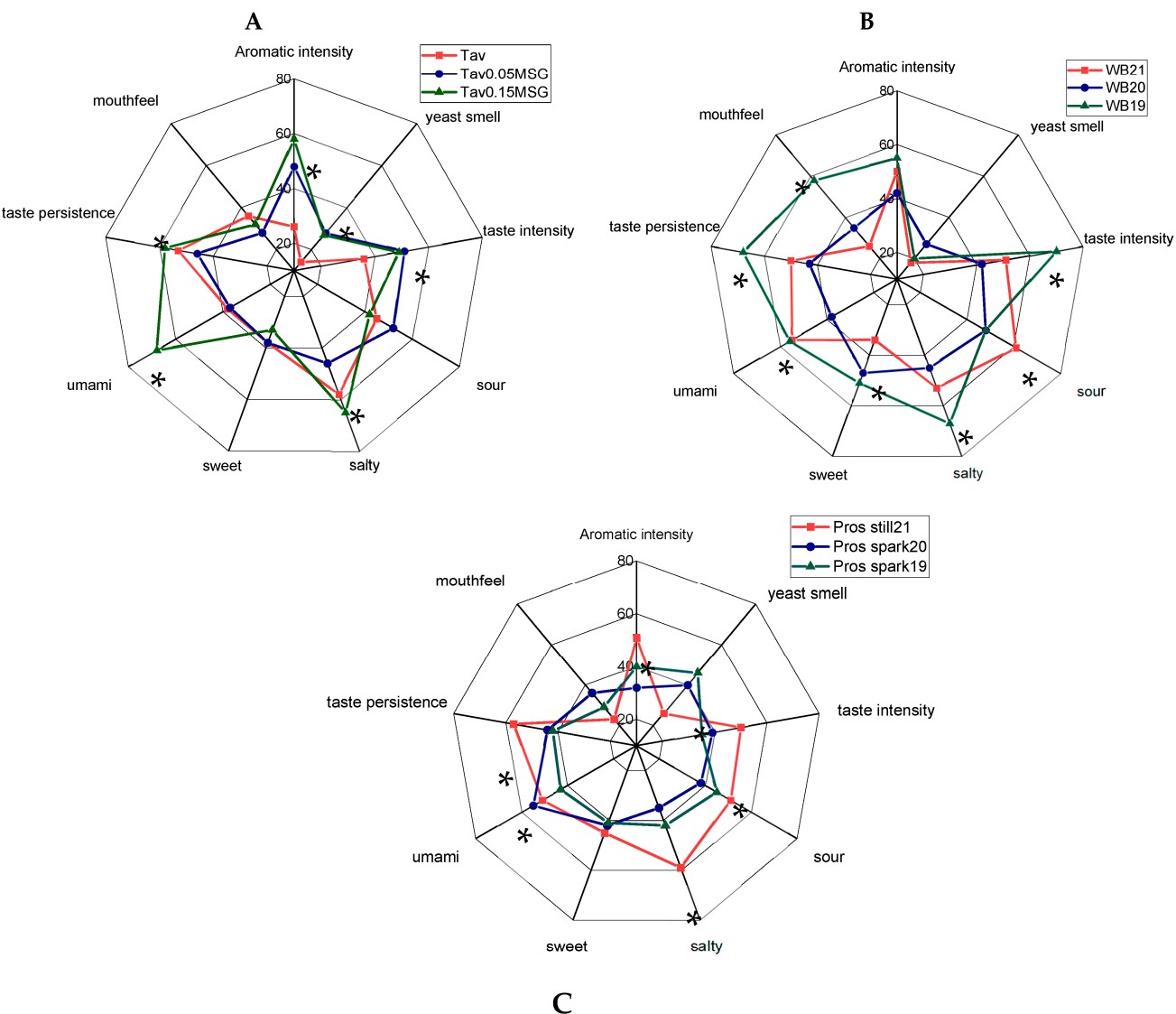

**Figure 1.** Spider plots of QDA analysis carried out on: (**A**) Tavernello wine without MSG (Tav sample), with 0.05 g/L MSG (Tav0.05MSG), and 0.15 g/L MSG (Tav0.15MSG). (**B**) Bianco Colli di Conegliano DOCG of 2019 (WB19), 2020 (WB20), and 2021 (WB21) vintage. (**C**) Vin col fondo Prosecco DOCG of 2019 (Pros spark19), 2020 (Pros spark20), and 2021 (Pros still21) vintage. Within the same descriptor, QDA scores marked with an asterisk (*) are significantly different from the others performed by ANOVA and Tukey's test ($p \leq 0.05$).

As shown in Figure 1B, umami was better perceived in WB19 and WB21, the sweet taste was more intense in WB20 and WB19, and the salty taste was higher in WB19 than in the other two wines. While the acidity was stronger in WB21, the taste intensity, mouthfeel and taste persistence were more marked in WB19. As shown in Figure 1B, it can be stated that the WB19 wine has a broader sensory profile than WB20 and WB21, perhaps because of the more prolonged aging process.

Another oenological technique, in which an amino acid enrichment could occur in the wine, is the prolonged contact with yeast lees after the end of fermentation It is well-known that, when a sparkling wine is produced in the traditional method, the autolysis of yeasts which happen in the bottle causes an accumulation of amino acids. In fact, after some time, the yeast cells at the bottom of the bottle, through an autolysis mechanism operated by enzymes, release nitrogenous compounds, including amino acids, polysaccharides, and glycoproteins, responsible for the sensory quality of the final wine [27]. On the contrary,

sparkling wines obtained in large vessels and with shorter yeast contact (Martinotti method) have a lower concentration of amino acids [9]. Some wines are obtained with the "sur lies" technique which consists in keeping the dead yeast cells inside the wine until it is consumed, without disgorgement.

Three wines, obtained with the *sur lies* winemaking technique, were compared by sensory analysis. For the comparison, 3 different vintages were considered: 2019, 2020, and 2021, the first two samples were sparkling (Pros spark20 and Pros spark19), and the last was "still" (Pros still21). As shown in Figure 1C, the 3 Prosecco *sur lies* wines have different profiles; in particular, the umami taste is more intense in Pros still21 and Pros spark20, while the yeast smell is more marked in Pros spark20 and Pros spark19. This last result could be attributed to the yeasts, which remained in the wines without being filtered. Furthermore, in "Prosecco col fondo", the yeast aroma values that increase proportionally with the years could be attributed precisely to the degradation of the yeast cell wall over time. However, the Pros spark19 showed the lowest value in terms of umami taste; this result, rather difficult to explain, could be due to the difficulty of recognizing/identifying this taste with its own identity.

Interestingly, Pros still21 shows higher values for the descriptors salty, taste intensity, aromatic intensity, and taste persistence; however, it was the base wine, still to be refermented in the bottle, with completely different aromas from the other two previous vintages. Thus, it is difficult to compare it to the other two vintages.

### 3.3. Principal Component Analysis

PC2, explaining 21.22% of the total variance, well describes the differences in the amount of MSG and Umami, which are directly correlated also with the other umami-related variables: yeast smell and ASP. The discrimination of PC2 towards umami is confirmed by the distribution of the Tav series, assigning negative values to Tav and Tav0.05MSG, while Tav0.15MSG has the highest positive values following the MSG and umami descriptors.

Interestingly, equal negative PC2 values were reported by WB20 (16 months of *batonnage*) and WB21 (2 months of *batonnage*). This indicates that, even if the *batonnage* time influenced the MSG content (43.27 mg/L in WB20; 24.76 mg/L in WB21), the perception of umami was not. WB21 and WB20 are also associated with high acidity. Considering that this indicator is in contrast with umami with respect to PC2, it could be hypothesized that the high acidity may decrease (or cover) the umami perception. This can also explain why also other samples with high MSG and ASP content, such as Tav 0.05MSG (97.16 and 46.60 mg/L, respectively), Pros still21 (45.64 and 45.62 mg/L, respectively), and Tav (47.13 and 44.25 mg/L, respectively), reported negative PC2 values. This umami/acidity contract was recently noted by other authors [28]. Among WB wines, the highest amounts of both MSG and ASP (58.76 and 53.55 mg/L, respectively) were detected for WB19 (12 months of *batonnage*), which, accordingly, reported the highest PC2 values of this series. This sample is also the one reporting the highest values of PC1, being perceived as the saltiest, tastiest, and most aromatic among all the tested wines.

PC1, explaining 40.85% of total variance, clearly shows that umami and MSG are correlated with global pleasantness, taste intensity, salty, aroma intensity, and taste persistence. Conversely, these sensory attributes are in contrast with the others related to umami (yeast smell and ASP). The link between these two latter factors may be represented by diacetyl, a volatile molecule produced during aspartate catabolism which, at low concentrations, is associated with different smells, including the one of yeast [29,30]. In this context, the low like score of Pros Spark 19 (2 years on lees) and especially Pros Spark 20 (1 year on lees) seems associated with high values of yeast smell and ASP, here linked to a reduced wine complexity.

Even if a correlation between aging on lees/*batonnage* time and umami-related sensory indicators was not found among these samples, it seems clear that high umami perception scores bring a more complex sensory profile and, consequently, a higher overall liking

of the product. This confirms that, even in wine where MSG values are generally below the detection threshold, different umami-related molecules impact the sensory profiles by enhancing the other tastes.

However, even if MSG is the molecule closest to the standard umami taste, its positive but weak correlation with taste and aroma complexity allows us to hypothesize that other umami-related molecules could better describe this taste-enhancing activity. In this context, possible candidates are yeast peptides and amino acids, while aspartate seems to only be associated with the standard umami taste and not the taste-enhancing activity.

### 3.4. Multiple Linear Regression

Since umami taste does not have a strong identity in wine but has the effect of taste enhancer, as reported by Klosse and colleagues [5], it could be defined by other sensory parameters, such as taste persistence.

By observing Figure 2 (PCA analysis), we decided to consider the parameter "taste persistence" as the expression of the taste-enhancing activity of umami. Therefore, the multiple linear regression of "taste persistence" in the function of other sensory parameters associated with taste and flavor enhancement was considered. Figure 3 shows the results of fitting a multiple linear regression model to describe the relationship between taste persistence and six independent variables. The equation of the fitted model is:

$$\text{taste persistence} = -5.27215 + 0.635248 \times \text{taste intensity} + 0.143442 \times \text{umami} + 0.259318 \times \text{yeast smell} - 0.10801 \times \text{mouthfeel} + 0.763276 \times \text{salty} - 0.530521 \times \text{Aromatic intensity}$$

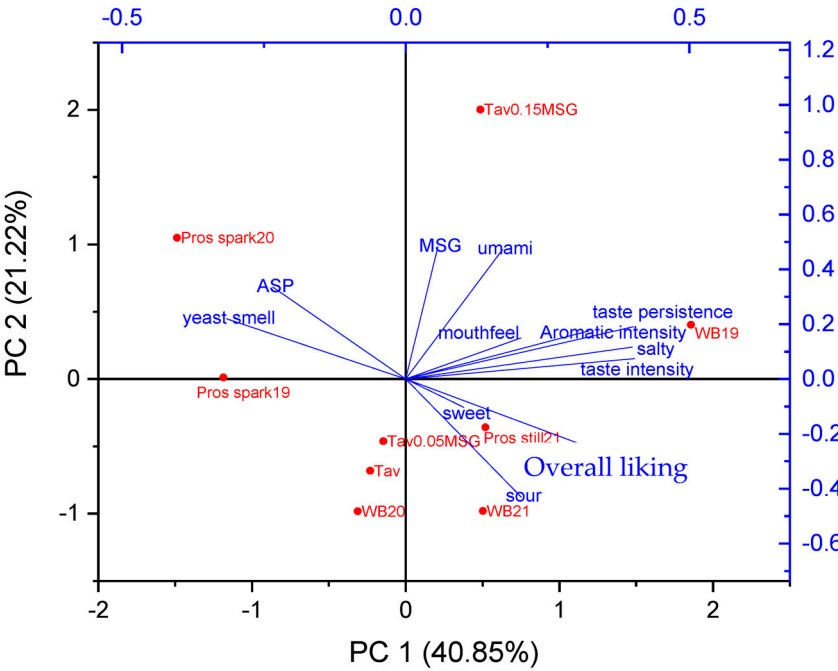

**Figure 2.** Principal component analysis (PCA) of sensory (QDA) and analytical parameters (glutamic acid and aspartic acid) detected in the analyzed wine samples.

Since the *p*-value in the ANOVA table is less than 0.05, there is a statistically significant relationship between the variables at the 95.0% confidence level. The R-squared statistic indicates that the fitted model explains 99.4678% of the variability in taste persistence. The six parameters, chosen for this model, are those that showed $p < 0.05$.

As shown in the function, in this linear regression, persistence is associated with those parameters that represent sapidity, intensity, and saltiness. Additionally, in this case the contribution of standard umami taste is clear but non prominent.

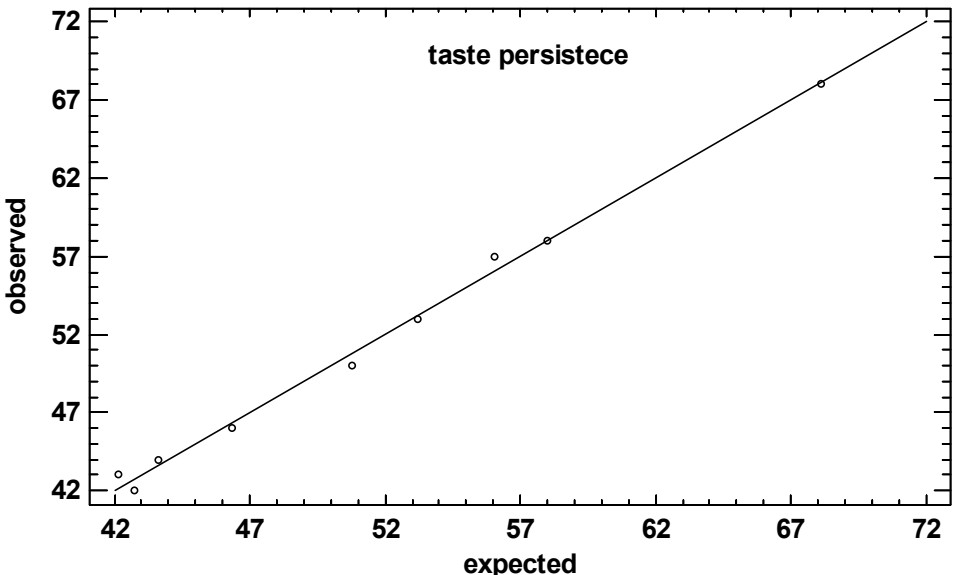

**Figure 3.** Multiple linear regression model to describe the relationship between taste persistence and six independent variables, expressed by score. The equation of the fitted model is: taste persistence = $-5.27215 + 0.635248 \times$ taste intensity $+ 0.143442 \times$ umami $+ 0.259318 \times$ yeast smell $- 0.10801 \times$ mouthfeel $+ 0.763276 \times$ salty $- 0.530521 \times$ Aromatic intensity. $R^2 = 0.995$, $p < 0.05$.

## 4. Conclusions

Wine is a drink with a very complex composition, made up of hundreds of molecules that contribute to its final flavor. The presence of umami in wine, even if documented, is little studied, also because it is difficult to identify and often associated with other sensory perceptions. In this research, it was necessary to get the panel familiar with the umami taste starting from an aqueous solution and, subsequently, adding monosodium glutamate and aspartate in neutral wines also to identify the perception thresholds in wine, which is a complex matrix. Thanks to the identification and quantification of umami in neutral white wines, it was possible to study this taste in sparkling wines and wines obtained by contact *sur lies*.

The sensory and analytical results did not show a strong correlation between the umami taste and the quantity of glutamate and aspartate, but their presence plays a fundamental role in enhancing other gustatory and olfactory perceptions with effects also on their persistence. The gustatory persistence is in linear regression ($R^2 = 0.99$), with those sensory parameters particularly associated with the sapidity of the wine. Apart from the widely studied monosodium glutamate, the presence of other umami-related molecules and their possible role as taste enhancers in wine needs to be further studied.

**Author Contributions:** D.F., methodology, G.L., formal analysis, writing, funding acquisition, R.S., investigation, original draft preparation, S.V., Conceptualization, software, validation, formal analysis, A.D.I., writing—review and editing, supervision. All authors have read and agreed to the published version of the manuscript.

**Funding:** This research was funded by the University of Padova Project prot. BIRD165379. And The Article Processing Charge was funded by the University of Padova Project prot. BIRD165379.

**Data Availability Statement:** Supplementary data are available upon reasonable request.

**Conflicts of Interest:** The authors declare no conflict of interest.

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
