# Peer review of "Umami in Wine: Impact of Glutamate Concentration and Contact with Lees on the Sensory Profile of Italian White Wines"

_beverages, doi:10.3390/beverages9020052_

Round 1

Reviewer 1 Report

Comments to the Author:

This study investigated whether glutamate concentration or contact with lees influenced the sensory profile of Italian white wines. The study is well-written, however, more details are needed in the materials and methods section. Other comments are listed below.

Line 92: Why is it essential to train panelists in tactile recognition? What does tactile perception mean specifically in this study?

Line 93: Please specify the test content regarding the threshold of umami and salty, e.g. what are the different concentrations of sodium glutamate and sodium chloride used in water.

Line 109: Please specify the use of only 9 panelists, and after the training, whether some panelists were eliminated based on the threshold test results.

Line 110: What were the environmental conditions of the sensory analysis room?

Line 123: In the gustatory-tactile recognition test, did the judges need to taste the solution in 8 glasses in turn, please specify. What is the six pure compounds for six perceptions? And how about their concentration?

Why did not use the 3-AFC test?

Line 167: Why was a scale (from 0 to100) used? The same below. And why is there no reference concentration provided, and that you may define its scale? Commonly, the 5-point scale and 9-point scale are used methods for strength scale.

Line 189-200: What was the flow rate?

Line 206: "p" should be italicized in "p<0.05"

Line 235: What does the "*" in Figure 1 stand for? Please specify in the figure legend.

What is the mouthfeel for white wine? How do you define this conception? And taste persistence.

Author Response

Dear Reviewer 

Below you will find the answers and clarifications to the doubts raised by your review.
Best regards

Reviewer 1

This study investigated whether glutamate concentration or contact with lees influenced the sensory profile of Italian white wines. The study is well-written, however, more details are needed in the materials and methods section. Other comments are listed below.

We thank very much the reviewer for the time taken to evaluate the manuscript and for the positive comments made.  All the highlighted points were fully considered below.

Line 92: Why is it essential to train panelists in tactile recognition? What does tactile perception mean specifically in this study?

In this research, tactile perception represents mouthfeel, which is the synergy among acidity, tannins, alcohol, glycerol, CO2 and other constituents also perceived at a tactile level. Since umami sensitizes more receptors, it was decided to include mouthfeel (tactile perception) to establish if there is any relationship between the two sensory parameters.

To better explain the "tactile perception" parameter we have modified the text as follow (Line 106): “the mouthfeel, intended as the synergy among acidity, tannins, alcohol, glycerol, CO2 and other constituents also perceived at a tactile level.

Line 93: Please specify the test content regarding the threshold of umami and salty, e.g. what are the different concentrations of sodium glutamate and sodium chloride used in water.

This test was fully explained in section 2.4 and the concentrations used are reported in Table 1. At line 116 (section “experimental design”) we have only mentioned it. To avoid misunderstanding, for each analysis mentioned in this paragraph, we added a reference to the section in which it is fully explained.

Line 109: Please specify the use of only 9 panelists, and after the training, whether some panelists were eliminated based on the threshold test results.

We have made changes to paragraph 2.1 and 2.2. We added as follows:

The analytical sensory panel, made up of 9 judges, is represented by highly trained people for more than 10 years on the evaluation of wine and the study of its parameters. It is regularly trained according to ISO standards, as specified in the text, and is the result of a selection that took place about 10 years ago.

The number of 9 judges is regularly admitted in sensory analysis, when the analytical panel has to conduct an analytical sensory test. In accordance with other protocols published in international articles, we used 9 judges. The training was repeated many times, in order to familiarize the judges with the descriptors, in this case with the umami taste in particular. Repeated training resulted in a trained panel of the same 9 people, without further selection of judges within the panel.

We added (Line 127): The analytical sensory panel, made up of 9 judges, is represented by highly trained people, for more years on the evaluation of wine and the study of its parameters. It is regularly trained according to ISO standards, and is the result of a selection that took place years ago. The training was repeated many times, to familiarize the judges with the descriptors, and with the umami taste in particular. The training made it possible to obtain a panel of 9 judges, prepared for the recognition of the descriptors, without further selections.

Line 110: What were the environmental conditions of the sensory analysis room?

Line 134: at room temperature (21 ± 0.5 °C) and artificial light

Line 123: In the gustatory-tactile recognition test, did the judges need to taste the solution in 8 glasses in turn, please specify. What is the six pure compounds for six perceptions? And how about their concentration?

Line 148: We added: During the first session, the judges had to distinguish six perceptions (sweet, salty, sour, bitter and umami and mouthfeel) in eight standards (eight different concentrations for each standard parameter associated to each perception), according to ISO 8586:2012 [14].

Why did not use the 3-AFC test?

The reviewer rightly suggests 3-AFC test. Although it would have been very useful in this case, it was not taken into consideration because we wanted to directly test the growth of each perception, linked to the ramp of increasing concentrations of the single descriptors.

Line 167: Why was a scale (from 0 to100) used? The same below. And why is there no reference concentration provided, and that you may define its scale? Commonly, the 5-point scale and 9-point scale are used methods for strength scale. .

As reported by the reviewer, we have not justified the use of the 0-100 scale. Indeed, the methodology proposed by the International Union of Oenologists IUOE was used. All parameters were assessed using the 100-point scale according to the International Union of Oenologists IUOE. Line 193-194, we added: “All parameters were assessed using the 100-point scale according to the International Union of Oenologists IUOE.

Line 189-200: What was the flow rate?

The flow rate (1 mL/min) was added in the text.

Line 206: "p" should be italicized in "p<0.05"

Corrected as suggested by the reviewer

Line 235: What does the "*" in Figure 1 stand for? Please specify in the figure legend.

The Figure’s caption has been rewritten to make it clearer and the meaning of asterisk has been now clarified.

What is the mouthfeel for white wine? How do you define this conception? And taste persistence.

Line (215): We added in the text: “Persistence is the permanence of olfactory-gustatory perceptions over time and is measured in seconds”.

“Mouthfeel”, as defined above in text of the manuscript, is the physical sensations due to the presence of chemical compounds, such as: acids, tannins, alcohol, glycerol, CO2 (L. 106-107).

Reviewer 2 Report

In my opinion, the manuscript "Umami in wine: impact of glutamate concentration and contact with lees on the sensory profile of Italian white wines" represents an interesting contribution to the research regarding the "umami synergy" effect.

-If we limit just to sipping a wine, we do not perceive umami taste, however by combining the wine with foods able to promote this synergy, the perception of umami can become very intense -.

Some recent studies have investigated this effect.

For this reason, the principal focus of manuscript should highlight more the role of umami taste and its influence of other taste-olfactory perceptions and their persistence. In my opinion, the training of panel to recognize the umami taste and verify its perception in a neutral wine with the addition of monosodium glutamate (lines 81-82) is secondary, and it does not add originality at the research (lines 81-87).

What is written in the in lines 26-37 is not clear. Could the authors specify better how the salts releasing a proton, they can elicit the umami taste?

Paragraph 2.1 (Material and Methods) is unclear. Three sessions are described in the experimental design, but it is not reported how many tests were carried out in each session. The tests carried out within each session, are those reported in the following paragraphs? If so, this information must be included in the text so to anticipate the description of sessions will take place later.

In addition, could the authors write which specific “tactile descriptor” has been considered in training the panel? (Just if a specific attribute was considered).

Finally, in this paragraph, the authors wrote “in the third session, a quantitative descriptive analysis (QDA) including different umami taste indicators was performed on the 3 neutral wines of the second session” but in the second session no mention to different wines but only one (Tavernello, Caviro, Italia). The authors could specify which is the second session because it is unclear (lines 95-99).

Table 2: The solution with concentration of 0.05 g/L of monosodium glutamate was not used in the test for the umami recognition threshold in water, described in the paragraph 2.4. However, this solution was chosen for the test described in the paragraph 2.5 also if the authors affirmed that "the solutions were prepared basing on the perception threshold in aqueous solution measured during the first sensory session". So, the choice of the solution at 0.05 g/L (lower concentration to 0.15 g/L, table 1) is not clear. Could the authors specify?

In addition, in the caption of table 2, specify the abbreviation Tav: tavernello wine.

Line 164-168. Why did the judges evaluate 4 samples while only 3 are shown and described in the table 2?

Lines 178-179: The authors should mention the fourth session also in the in the experimental design (paragraph 2.1).

Line 186: The panel has been trained on the sensory attributes request in the study. However, the authors no mention the training of panel regarding the yeast smell attribute. According to the authors, may there be negative interaction between umami and yeast smell? Could the presence of the yeast smell because decreasing of the perception of umami taste? For this reason, could be important a specific training of panel regarding the yeast smell? Perhaps, the panel were experts regards to this attribute.

Line 212. The reference 14 does not seem to refer to this specific topic. I recommend to authors to control.

Lines 232-234: In the other Tav wine samples, added with MSG, the panelists recognized 100% the umami taste, when present in concentrations of 0.097 g/L in Tav0.05MSG and 0.181 g/L in 233 Tav0.15MSG, respectively. The authors did not consider that the result could be due to the affirmation written in the lines 165-166 (At the beginning of the test the judges were informed that the solutions had been prepared with the compound responsible for the umami taste).  Could this information have influenced the judges?

Figure 1. I suggest evidence more the lines that represent the samples because they are confused with the lines of radar chart.

Lines 282-286. Have the authors considered that results could depend also to the grape variety as the authors reported in introduction (lines 48-49)?

Line 338 and line 352. The authors written "blue circle and red circle" but these are not represented in the figure.

In my opinion, the authors could add in the "results and discussion" a table of the results regarding ASP and MSG content.

Author Response

Dear Reviewer

Below you will find the answers and clarifications to the doubts raised by your review.
Best regards

Reviewer 2

In my opinion, the manuscript "Umami in wine: impact of glutamate concentration and contact with lees on the sensory profile of Italian white wines" represents an interesting contribution to the research regarding the "umami synergy" effect.

We thank very much the reviewer for the time taken to evaluate the manuscript and for the positive comments made.  All the highlighted points were fully considered below.

-If we limit just to sipping a wine, we do not perceive umami taste, however by combining the wine with foods able to promote this synergy, the perception of umami can become very intense -.

Some recent studies have investigated this effect.

For this reason, the principal focus of manuscript should highlight more the role of umami taste and its influence of other taste-olfactory perceptions and their persistence. In my opinion, the training of panel to recognize the umami taste and verify its perception in a neutral wine with the addition of monosodium glutamate (lines 81-82) is secondary, and it does not add originality at the research (lines 81-87).

The reviewer highlights an important point of the manuscript. But, during the experimentation, problems emerged regarding the identification/recognition of umami, especially in a matrix, such as wine, in which it is little studied. For this reason, we wanted to study the umami taste in an aqueous solution and in a neutral wine to understand, right from the beginning of the experiment, the relationship on its detection and the influence with other descriptors. This initial setting of the experimentation has become necessary to study the subsequent phases, even if it may seem devoid of originality.

What is written in the in lines 26-37 is not clear. Could the authors specify better how the salts releasing a proton, they can elicit the umami taste?   

Since the explanation, requested by the reviewer, required an in-depth discussion on the physiological aspect of umami which are not really the main objective of this research, we preferred to eliminate the sentence "In solution, this salt releases a proton capable of eliciting the umami taste".

Paragraph 2.1 (Material and Methods) is unclear. Three sessions are described in the experimental design, but it is not reported how many tests were carried out in each session. The tests carried out within each session, are those reported in the following paragraphs? If so, this information must be included in the text so to anticipate the description of sessions will take place later.

Yes, paragraph 2.1 (“Experimental design”) only includes an overview of the analysis conducted in this study. To avoid misunderstanding, for each analysis mentioned in this paragraph, we added a reference to the section in which it is fully explained.

In addition, could the authors write which specific “tactile descriptor” has been considered in training the panel? (Just if a specific attribute was considered).

This observation was also made by another reviewer because, indeed, as reported in the manuscript, the "tactile perception" is explained in an unclear way. For this reason, we explain the descriptor as follows:

In this research, tactile perception represents mouthfeel, which is the synergy among acidity, tannins, alcohol, glycerol, CO2 and other constituents also perceived at a tactile level. Since umami sensitizes more receptors, it was decided to include mouthfeel (tactile perception) to establish if there is any relationship between the two sensory parameters.

To better explain the "tactile perception" parameter we have modified the text as follow (Line 106): “the mouthfeel, intended as the synergy among acidity, tannins, alcohol, glycerol, CO2 and other constituents also perceived at a tactile level.

Finally, in this paragraph, the authors wrote “in the third session, a quantitative descriptive analysis (QDA) including different umami taste indicators was performed on the 3 neutral wines of the second session” but in the second session no mention to different wines but only one (Tavernello, Caviro, Italia). The authors could specify which is the second session because it is unclear (lines 95-99).

The reviewer is right, it was unclear. We now revised the paragraph to avoid misinterpretations (102-122).

Table 2: The solution with concentration of 0.05 g/L of monosodium glutamate was not used in the test for the umami recognition threshold in water, described in the paragraph 2.4. However, this solution was chosen for the test described in the paragraph 2.5 also if the authors affirmed that "the solutions were prepared basing on the perception threshold in aqueous solution measured during the first sensory session". So, the choice of the solution at 0.05 g/L (lower concentration to 0.15 g/L, table 1) is not clear. Could the authors specify?

In the neutral wine, during the training phase of the panel for the recognition of the threshold, it was decided to add a lower quantity of monosodium glutamate (0.05 g/L), compared to the aqueous solutions of Tab. 2, because monosodium glutamate is already a molecule present in wine. Therefore, the wine already has a quantity of monosodium glutamate and it was decided to add a smaller quantity than the aqueous solution.

In addition, in the caption of table 2, specify the abbreviation Tav: tavernello wine.

Corrected as suggested by the reviewer.

Line 164-168. Why did the judges evaluate 4 samples while only 3 are shown and described in the table 2?

The sentence was wrongly written. Samples were 3 as described in Table 2. We now corrected the error.

Lines 178-179: The authors should mention the fourth session also in the in the experimental design (paragraph 2.1).

This point was addressed by entirely revising the experimental design paragraph.  

Line 186: The panel has been trained on the sensory attributes request in the study. However, the authors no mention the training of panel regarding the yeast smell attribute. According to the authors, may there be negative interaction between umami and yeast smell? Could the presence of the yeast smell because decreasing of the perception of umami taste? For this reason, could be important a specific training of panel regarding the yeast smell? Perhaps, the panel were experts regards to this attribute.

The panel was expert on the yeasty smell. In fact, also in other trainings, bakery yeast or wine lees were used to recognize the smell of yeast.

We added (Line 213-214): “the smell of commercial bakery yeast”

Line 212. The reference 14 does not seem to refer to this specific topic. I recommend to authors to control.

The reviewer is right. We replaced reference 14 with references 22 and 23 in this point.

Lines 232-234: In the other Tav wine samples, added with MSG, the panelists recognized 100% the umami taste, when present in concentrations of 0.097 g/L in Tav0.05MSG and 0.181 g/L in 233 Tav0.15MSG, respectively. The authors did not consider that the result could be due to the affirmation written in the lines 165-166 (At the beginning of the test the judges were informed that the solutions had been prepared with the compound responsible for the umami taste).  Could this information have influenced the judges?

As rightly noted by the reviewer, the sentence may have influenced the judges. But they were only informed about the type of study they were conducting, with the crucial part of the work being the quantification and detection of the umami threshold in blinded samples.

Figure 1. I suggest evidence more the lines that represent the samples because they are confused with the lines of radar chart.

We have presented a new version of Fig. 1

Lines 282-286. Have the authors considered that results could depend also to the grape variety as the authors reported in introduction (lines 48-49)?

Yes, we have. For this reason, we choose different wines and samples’ data were, in this chapter, discussed within the same wine series (Tav, Pros, WB). To address the reviewer’s comment and remark the importance of grape variety, a sentence has been modified (L 241, 242).

Line 338 and line 352. The authors written "blue circle and red circle" but these are not represented in the figure.

Apologies to the reviewer, but "blue circle and red circle" were a mistake. We have eliminated the two elements from the text.

In my opinion, the authors could add in the "results and discussion" a table of the results regarding ASP and MSG content.

These data were already included in the dataset used to prepare Figure 2 and mentioned in the text when needed. Therefore, we believe that it is not necessary to report them also in an additional table. Consequently, no action was taken regarding this point.

Reviewer 3 Report

I was tasked to evaluate the manuscript "Umami in wine: impact of glutamate concentration and contact with lees on the sensory profile of Italian white wines". The paper is well written, clear and focused on an interesting topic with practical applications. No self citation was detected and the english form is adequate. I really appreciated the experimental design and also the multivariate statistical approach. I just have some comments before the pubblication. 

Line 36: Please add a reference to support this statement. I suggest also to better explain this sensory mechanism if it is possible.

Line 41: the relationship between umami taste and “inosinate, and guanylate” should be supported by a dedicated reference.

Line 53: I suggest to replace “main” with “most abundant” or other words with the same meaning.

Line 72: the authors gave an overview of the presence of umami taste in several different winemaking styles but the influence of maceration on it is missing. Even a few words regarding the presence of this taste in white and red wines as well in passito – like wines is welcome if any data is available in literature.

Line 75: this statement should be supported by a reference. The concept of “overall sensory perception” needs a brief explanation.

Line 78: check for the typo “-“ before “Furthermore”.

Line 129: why the reduced amount of sodium was crucial for this purpose? What is the average concentration of this cation in wines?

Table 2: this table is not clear and must be modified. I was not able to identify the 4 samples tested by panelists.

Line 209 – 221: this part is more coherent to the introduction and this information were introduced before. I invite the authors to modify this introductory part of the R&D section to make it more concise.

Line 231: this is very interesting. Any correlation with other taste enhancers such as methoxypyrazines?

Section 3.2. how do the authors evaluate the effect of MSG? Because over 2 years MSG accumulated but also all other compounds content changed..

Line 297: The authors must uniform “sur lies” or “sour lees” as used in the introduction.

Did the author evaluate the concentration of salts? A general non-specific value related to salt concentration such as conductivity could be a good parameter to estimate the salting out effect which is probably somehow related to umami.

English is fine and adequate.

Author Response

Dear Reviewer

Below you will find the answers and clarifications to the doubts raised by your review.
Best regards

Reviewer 3

I was tasked to evaluate the manuscript "Umami in wine: impact of glutamate concentration and contact with lees on the sensory profile of Italian white wines". The paper is well written, clear and focused on an interesting topic with practical applications. No self citation was detected and the english form is adequate. I really appreciated the experimental design and also the multivariate statistical approach. I just have some comments before the pubblication. 

We thank very much the reviewer for the time taken to evaluate the manuscript and for the positive comments made.  All the highlighted points were fully considered below.

Line 36: Please add a reference to support this statement. I suggest also to better explain this sensory mechanism if it is possible.

Since the explanation, requested by the reviewer, required an in-depth discussion on the physiological aspect of umami and, since the physiological aspect is not really the main objective of this research, we preferred to eliminate the sentence "In solution, this salt releases a proton capable of eliciting the umami taste".

Line 41: the relationship between umami taste and “inosinate, and guanylate” should be supported by a dedicated reference.

We added a new reference addressing this point. (L39-40)

Line 53: I suggest to replace “main” with “most abundant” or other words with the same meaning.

Corrected as suggested by the reviewer

Line 72: the authors gave an overview of the presence of umami taste in several different winemaking styles but the influence of maceration on it is missing. Even a few words regarding the presence of this taste in white and red wines as well in passito – like wines is welcome if any data is available in literature.

No papers have been published on the effect of winemaking technologies on the expression of umami taste, however as the umami depends mainly on the content of glutammic and aspartic acid, some hypothesis can be drawn. To address this point a paragraph has been added (57-66)

Line 75: this statement should be supported by a reference. The concept of “overall sensory perception” needs a brief explanation.

The reviewer is right. Therefore, the sentence has been rewritten and a reference has been added to support the statement (L85-87).

Line 78: check for the typo “-“ before “Furthermore”.

The symbol is not present now.

Line 129: why the reduced amount of sodium was crucial for this purpose? What is the average concentration of this cation in wines?

In this experiment we used the water we always use for sensory analysis. We would like to point out that the low concentration of sodium is a feature shown on the water label. However, we did not want to emphasize the low sodium concentration, but only underline that we choose a commercial water with a very low mineral content (probably the lowest in the Italian market) as it represents the matrix closest to pure water among commercial products.

Table 2: this table is not clear and must be modified. I was not able to identify the 4 samples tested by panelists.

The reviewer is right. There was an error at line 168: samples were 3, not four. The table was also modified to be clearer.

Line 209 – 221: this part is more coherent to the introduction and this information were introduced before. I invite the authors to modify this introductory part of the R&D section to make it more concise.

The paragraph has been revised and made more concise (L 238-244).

Line 231: this is very interesting. Any correlation with other taste enhancers such as methoxypyrazines?

Unfortunately, in this research we only have information about glutamate and aspartate content while other taste enhancers were not considered.

Section 3.2. how do the authors evaluate the effect of MSG? Because over 2 years MSG accumulated but also all other compounds content changed.

In this research it was hypothesized that after two years MSG may have accumulated in the wine. there is no doubt that other compounds can also be changed; but we hypothesized only the effect of the potential accumulation of MSG following the presence of yeasts.

Line 297: The authors must uniform “sur lies” or “sour lees” as used in the introduction.

We replaced “sur lees” with “sur lies” whenever present in the text.

Did the author evaluate the concentration of salts? A general non-specific value related to salt concentration such as conductivity could be a good parameter to estimate the salting out effect which is probably somehow related to umami.

Unfortunately, this analysis has not been conducted in this study. However, the molecules responsible for the direct (umami taste) and indirect (other tastes’ modulation) umami effects (gltammate and aspartate) have been quantified and included  in the dataset of Figure 2.

Round 2

Reviewer 1 Report

Accept

Reviewer 2 Report

Related the Author's Notes:

A)  In the neutral wine, during the training phase of the panel for the recognition of the threshold, it was decided to add a lower quantity of monosodium glutamate (0.05 g/L), compared to the aqueous solutions of Tab. 2, because monosodium glutamate is already a molecule present in wine. Therefore, the wine already has a quantity of monosodium glutamate and it was decided to add a smaller quantity than the aqueous solution.

 R) I suggest that the authors report this consideration inside the manuscript.

A)  The panel was expert on the yeasty smell. In fact, also in other trainings, bakery yeast or wine lees were used to recognize the smell of yeast.

R)     This information does not evince in the text, I suggest reporting it.

A)   As rightly noted by the reviewer, the sentence may have influenced the judges. But they were only informed about the type of study they were conducting, with the crucial part of the work being the quantification and detection of the umami threshold in blinded samples.

R)  Considering what the authors already wrote, I believe that the following sentence (in paragraph 2.5, now lines 191-192) misleading: "At the beginning of the test the judges were informed that the solutions had been prepared with the compound responsible for the umami taste".  The sentence interpretation suggest that authors informed the judges about la presence of umami taste in the solution prepared for "umami recognition threshold in wine". It is evident that the information causes an "anticipation responses error". For this reason, I think is better to avoid the sentence.
